# Biomass burning pollution in the South Atlantic upper troposphere: GLORIA trace gas observations and evaluation of the CAMS model

Sören Johansson[1], Gerald Wetzel[1], Felix Friedl-Vallon[1], Norbert Glatthor[1], Michael Höpfner[1], Anne Kleinert[1], Tom Neubert[2], Björn-Martin Sinnhuber[1], and Jörn Ungermann[3]

[1]Institute of Meteorology and Climate Research - Atmospheric Trace Gases and Remote Sensing (IMK-ASF), Karlsruhe Institute of Technology, Karlsruhe, Germany
[2]Central Institute of Engineering, Electronics and Analytics - Electronic Systems (ZEA-2), Forschungszentrum Jülich, Jülich, Germany
[3]Institute of Energy and Climate Research - Stratosphere (IEK-7), Forschungszentrum Jülich, Jülich, Germany

**Correspondence:** S. Johansson (soeren.johansson@kit.edu)

**Abstract.** In this study, we present simultaneous airborne measurements of peroxyacetyl nitrate (PAN), ethane ($C_2H_6$), formic acid (HCOOH), methanol ($CH_3OH$), and ethylene ($C_2H_4$) above the South Atlantic in September and October 2019. Observations were obtained from the Gimballed Limb Observer for Radiance Imaging of the Atmosphere (GLORIA), as two-dimensional altitude cross-sections along the flight path. The flights were part of the SouthTRAC (Transport and Composition in the Southern Hemisphere Upper Troposphere/Lower Stratosphere) campaign with the German High Altitude and Long range research Aircraft (HALO). On two flights (8 September 2019 and 7 October 2019), large enhancements of all these substances were found between 7 and 14 km altitude with maximum volume mixing ratios (VMRs) of 1000 pptv of PAN, 1400 pptv for $C_2H_6$, 800 pptv for HCOOH, 4500 pptv for $CH_3OH$, and 200 pptv for $C_2H_4$. One flight shows a common filamentary structure in the trace gas distributions, while the second flight is characterized by one large plume. Using backward trajectories, we show that measured pollutants likely reached UTLS altitudes above South America and central Africa, where elevated PAN VMRs are visible at the surface layer of the Copernicus Atmosphere Monitoring Service (CAMS) model during the weeks before both measurements. In comparison to results of the CAMS reanalysis interpolated onto the GLORIA measurement geolocations, we show that the model is able to reproduce the overall structure of the measured pollution trace gas distributions. For PAN, the absolute VMRs are in agreement with the GLORIA measurements. However, $C_2H_6$ and HCOOH are generally underestimated by the model, while $CH_3OH$ and $C_2H_4$, the species with the shortest atmospheric lifetimes of the pollution trace gases discussed, are overestimated by CAMS. The good agreement between model and observations for PAN suggests that the general transport pathways and emissions locations are well captured by the model. The poorer agreement for other species is therefore most likely linked to model deficiencies in the representation of loss processes and emission strength.

## 1 Introduction

The composition of the upper troposphere and lower stratosphere (UTLS) above South America and even above remote regions of the South Atlantic is strongly influenced by frequent biomass burning events in central Africa, South America, and

Australia (e.g., Andreae and Merlet, 2001; Glatthor et al., 2015). Pollution trace gases and aerosols, resulting from fires, can be transported to the UTLS. Here, some of these trace gases (e.g., carbon monoxide (CO), nitrogen dioxide ($NO_2$)) are able to produce ozone ($O_3$), which acts as efficient greenhouse gas at these altitudes (Xia et al., 2017; Bozem et al., 2017; Bourgeois

et al., 2021). Further, some pollution trace gases (in particular volatile organic compounds) may act as aerosol precursor (e.g., Hobbs et al., 2003; Lim et al., 2019; Akherati et al., 2020, and references therein). Together with directly emitted aerosols, they influence surface climate. It is well known that the UTLS is of great importance for global climate (Gettelman et al., 2011; Riese et al., 2012), and it is thus important to study biomass burning trace gases at UTLS altitudes. Due to increasing biomass burning activities in Africa, South America and Australia (Torres et al., 2010; Abram et al., 2021), the potential influence of

biomass burning trace gases on climate may increase over time.

Typical biomass burning trace gases have different atmospheric lifetimes and atmospheric sinks. Further, they may have additional non-pyrogenic sources, such as fuel combustion or biogenic emissions. In this work, peroxyacetyl nitrate (PAN), ethane ($C_2H_6$), formic acid (HCOOH), methanol ($CH_3OH$), and ethylene ($C_2H_4$) are discussed, and their characteristics are summarized in Tab. 1. These trace gases have been selected for this study, because they all are potentially emitted from biomass

burning events, since they have a large range of upper tropospheric lifetimes (from a few days to several months), and because they are part of the GLORIA (Gimballed Limb Observer for Radiance Imaging of the Atmosphere) measurements and of the CAMS (Copernicus Atmosphere Monitoring Service) model output. In addition, these trace gases are measured by various infra red satellite sounders (see below) but in coarser spatial resolution than the GLORIA measurements. Typical atmospheric sinks of those pollutants are reactions with the hydroxyl radical (OH), wet and dry deposition, and thermal decomposition

(see Tab. 1 for details). In previous studies, these trace gases have been observed by several satellite instruments in nadir (e.g., Coheur et al., 2009; Dolan et al., 2016; Franco et al., 2018; Pope et al., 2021) as well as in limb views (e.g., Rinsland et al., 2005; Dufour et al., 2007; Grutter et al., 2010; Wiegele et al., 2012). These measurements allowed for monitoring of large scale biomass burning plume transport and chemical composition. Further, these satellite measurements allowed to better estimate biomass burning sources (e.g., Stavrakou et al., 2011), and examined transport and composition of simulated biomass burning

plumes (e.g., Glatthor et al., 2013). In addition, airborne in situ (e.g., Singh et al., 2001; Peischl et al., 2018) and remote sensing (e.g., Ungermann et al., 2013; Johansson et al., 2020; Wetzel et al., 2021) observations provided detailed studies of filamentary structures (i.e. mesoscale structures with horizontal extension of up to hundreds of kilometers) of these pollution trace gases.

Atmospheric model simulation of such pollution trace gases is challenging: For good model performance, knowledge about

pollutant emissions, chemistry and transport are necessary. Location, time and emitted species of biomass burning events are typically represented by emission data sets in atmospheric models. If major fire events are missing in these emissions, the model is not able to reproduce upper tropospheric distributions of those species. Further, injection heights of polluted air masses from fires are difficult to estimate, and only few emission data sets include this information (e.g., Rémy et al., 2017). In addition, it is difficult to estimate the type of fuel of biomass burning events, and thus the knowledge of the concentration of

the emitted compounds is uncertain (e.g., van der Werf et al., 2017). Another challenge is the accurate simulation of transport pathways of polluted air masses during pyroconvective updraft (e.g., Khaykin et al., 2020). Furthermore, chemical reactions

**Table 1.** Sources, sinks, estimated lifetimes, and approximate upper tropospheric background volume mixing ratios (VMRs) of pollution trace gases PAN, $C_2H_6$, HCOOH, $CH_3OH$, and $C_2H_4$ which are discussed in this paper.

| | Sources | Sinks | Lifetime | Background VMRs | References |
|---|---|---|---|---|---|
| PAN | Precursors: | Thermal decomposition | 1 h (BL) | <100 pptv | Fischer et al. (2014) |
| | Fuel combustion | Photolysis (UT) | 5 months (UT) | | Glatthor et al. (2007) |
| | Biomass burning | | | | |
| $C_2H_6$ | Biomass burning | Reaction with OH | 2 months | <500 pptv (NH) | Rudolph (1995) |
| | Natural gas loss | | | <300 pptv (SH) | Xiao et al. (2008) |
| | Fossil fuel production | | | | |
| | Bio fuel use | | | | |
| HCOOH | Biogenic emissions | Reaction with OH | 1-2 days (BL) | <100 pptv | Paulot et al. (2011) |
| | Biomass burning | Wet and dry deposition | weeks (UT) | | Millet et al. (2015) |
| | Fossil fuel combustion | | | | Mungall et al. (2018) |
| | Secondary production | | | | |
| $CH_3OH$ | Biogenic emissions | Reaction with OH | 5.3 days | <300 pptv | Bates et al. (2021) |
| | Oceanic emissions | Wet and dry deposition | | | |
| | Biomass burning | Ocean uptake | | | |
| | Anthropogenic emissions | | | | |
| | Secondary production | | | | |
| $C_2H_4$ | Biogenic emissions | Reaction with OH | 0.5 days (LT) | <50 pptv | Sawada and Totsuka (1986) |
| | Biomass burning | Reaction with $O_3$ | 1.2 days (UT) | | Mauzerall et al. (1998) |
| | Fossil fuel combustion | | | | Morgott (2015) |

BL - boundary layer; LT - lower troposphere; UT - upper troposphere; NH - northern hemisphere; SH - southern hemisphere

and physical processes that are relevant for these biomass burning pollution trace gases are subject of recent research, such as the formation of HCOOH (Franco et al., 2021), or secondary aerosol formation (Lim et al., 2019). As an example, Wetzel et al. (2021) showed that it is difficult to reproduce measured pollution trace gas plumes above the North Atlantic without artificially

enhanced emissions. Earlier, Coheur et al. (2007) compared ACE-FTS observations of a biomass burning plume with their model and found considerable differences in the vertical profiles of CO.

Between September and November 2019, the SouthTRAC (Transport and Composition in the Southern Hemisphere Upper Troposphere/Lower Stratosphere; see https://www.pa.op.dlr.de/southtrac/, last access 17 January 2022) aircraft campaign was conducted with base in Rio Grande, Argentina. Onboard the German research aircraft HALO (High Altitude and Long range

research aircraft), ten in situ and three remote sensing instruments performed measurements during transfer flights from Oberpfaffenhofen, Germany to Rio Grande, Argentina, and during local flights from Rio Grande. The GLORIA instrument was part of the HALO payload and measured two dimensional distributions of temperature and various trace gases, such as PAN, $C_2H_6$, HCOOH, $CH_3OH$, and $C_2H_4$, along the flight track. In this work, we use these measurements to identify polluted air masses,

estimate the points at which the air masses reached UTLS altitudes using backward trajectories, and perform quantitative comparisons of the measurements with the CAMS model simulation results. The CAMS reanalysis uses a state-of-the-art atmospheric chemistry model for data assimilation, which is publicly available and widely used for air quality and pollution related studies (e.g., studies citing Inness et al., 2019). In this work, we aim to evaluate the CAMS reanalysis in the remote upper troposphere above the South Atlantic, a sparsely measured region. With our comparisons we further aim to provide recommendations for improving the CAMS model with respect to emissions and atmospheric lifetimes for the studied species.

In the following, we present an overview of GLORIA measurements and retrievals during the SouthTRAC campaign, together with a description of the atmospheric models used by this study. The measured concentrations of biomass burning trace gases for two research flights above the South Atlantic are discussed in detail, followed by a backward trajectory analysis to estimate the point at which these biomass burning plumes reached UTLS altitudes. Finally, GLORIA observations are directly compared to CAMS model simulation results.

## 2 Observations and atmospheric model simulations

### 2.1 GLORIA measurements during the SouthTRAC aircraft campaign

The GLORIA instrument (Friedl-Vallon et al., 2014; Riese et al., 2014) has been deployed on various aircraft campaigns with the Russian M55 Geophysica (e.g., Höpfner et al., 2019) and the German HALO research aircraft (e.g., Oelhaf et al., 2019). The instrument combines a Fourier-Transform-Spectrometer with an imaging detector, which allows for simultaneous observations of $128 \times 48$ atmospheric spectra. An actively controlled gimbal frame enables compensation of aircraft movements and targeted line of sight control. Two external black bodies as well as measurements into deep space are used for in-flight radiometric calibration. In this study, we use interferograms recorded in GLORIA's high-spectral-resolution mode. This mode utilizes an optical path difference of 8.0 cm, which results in spectral sampling of 0.0625 cm$^{-1}$, and a horizontal sampling along the flight path of approximately 3.5 km. These recorded interferograms are radiometrically and spectrally calibrated according to GLORIA level 1 processing as described by Kleinert et al. (2014), Guggenmoser et al. (2015), and Ungermann et al. (2021). The spectra are horizontally binned after filtering of bad pixels, such that the final data product is a set of 127 vertical spectra per measurement (one row is filtered completely).

These calibrated spectra are then used to retrieve atmospheric profiles of temperature, aerosols and trace gases. The retrieval applies a nonlinear least-squares fit algorithm with Tikhonov regularization, and is based on the retrieval strategy as described by Johansson et al. (2018). Overall, the retrievals are similar to retrievals of pollution trace gases as described by Wetzel et al. (2021) and Johansson et al. (2020). Specific retrieval properties are summarized in Tab. 2. Detailed error estimation and vertical resolution analyses are provided as supplement to this paper, and a summary per species is presented in Tab. 2. Spectroscopic errors are noted in the supplement and are, among other sources, based on uncertainties reported by Rothman et al. (2005) and Gordon et al. (2017).

GLORIA observations used in this study were recorded during the SouthTRAC HALO aircraft campaign. This campaign was conducted in two phases between September and November 2019, of which the phase in September focused on mea-

**Table 2.** Retrieval properties for PAN, $C_2H_6$, HCOOH, $CH_3OH$, and $C_2H_4$: Spectral regions used, and handling of interfering species. 10 and 90 percentile ranges are given for vertical resolution and estimated errors (combination of random and systematic errors). In the supplement, it is shown that larger absolute errors are typically connected to higher VMRs.

| Target gas | Spectral regions | Fitted species | Forward-calculated species | Vert. resolution | Estimated error |
|---|---|---|---|---|---|
| PAN | 780.0 - 790.0 cm$^{-1}$ <br> 794.0 - 805.0 cm$^{-1}$ | $H_2O^\dagger$, $O_3^\dagger$, HCFC-22, $CCl_4$, PAN | $CO_2$, $NO_2$, $NH_3$, $HNO_3^\dagger$, ClO, OCS, HCN, $CH_3Cl$, $C_2H_2$, $C_2H_6$, $COF_2$, $C_2H_4$, CFC-11$^\dagger$, CFC-113, CFC-141, $ClONO_2^\dagger$, $CH_3OH$, $HNO_4$, $CH_3COCH_3$ | 0.4 - 0.8 km | 40 - 130 pptv |
| $C_2H_6$ | 819.000 - 822.625 cm$^{-1}$ <br> 829.750 - 833.125 cm$^{-1}$ | $O_3^\dagger$, $C_2H_6$, HCFC-22 | $H_2O^\dagger$, $CO_2$, $NO_2$, $NH_3$, $HNO_3^\dagger$, HCN, $CH_3Cl$, $C_2H_2$, $COF_2$, CFC-11$^\dagger$, $CCl_4$, $ClONO_2^\dagger$, $CH_3CCl_3$, PAN | 0.8 - 1.1 km | 130 - 380 pptv |
| HCOOH | 1086.50 - 1089.44 cm$^{-1}$ <br> 1103.50 - 1106.12 cm$^{-1}$ <br> 1112.50 - 1116.88 cm$^{-1}$ | HCOOH, $O_3^\dagger$, CFC-12$^\dagger$, HCFC-22 | $H_2O^\dagger$, $CO_2$, $NH_3$, CFC-11$^\dagger$, CFC-113, CFC-114 | 0.5 - 1.2 km | 30 - 140 pptv |
| $CH_3OH$ | 982.875 - 999.312 cm$^{-1}$ | $H_2O^\dagger$, $O_3^\dagger$, $NH_3$, $CH_3OH$, PAN | $CO_2$, CFC-113 | 0.4 - 0.9 km | 200 - 680 pptv |
| $C_2H_4$ | 945.188 - 952.312 cm$^{-1}$ | $H_2O^\dagger$, $NH_3$, $C_2H_4$, $SF_6$ | $CO_2$, $O_3^\dagger$, $N_2O$, $NO_2$, $HNO_3^\dagger$, $COF_2$, CFC-11$^\dagger$, CFC-12$^\dagger$, HCFC-22, $CH_3CCl_3$, $CH_3COCH_3$, PAN | 0.3 - 0.7 km | 30 - 110 pptv |

$^\dagger$ Results of previous retrievals (not shown in this paper) targeting these species have been used for simulation of the spectra, or as an initial guess for the retrieval.

surements of gravity waves (Rapp et al., 2021). GLORIA measurements have been obtained during transfer flights between Oberpfaffenhofen, Germany, and Rio Grande, Argentina, and local flights from Rio Grande. In this study, results from two transfer research flights are discussed. The flights on 8 September 2019 and 7 October 2019 were directed from Sal, Cape Verde to Buenos Aires, Argentina, and vice versa, as part of the aircraft transfer between Germany and Rio Grande. Those flights have been selected for this study, because they covered the UTLS region above the South Atlantic and revealed the highest volume mixing ratios of the examined pollution trace gases.

## 2.2 CAMS atmospheric model

The Copernicus Atmosphere Monitoring Service (CAMS) provides the ECMWF (European Centre for Medium-Range Weather Forecast) Atmospheric Composition Reanalysis version 4 (EAC4) data product (Flemming et al., 2015; Inness et al., 2019). This data product (in this manuscript abbreviated as "CAMS") utilizes the ECMWF Integrated Forecast System (IFS) model, which assimilates various observations of atmospheric state and composition. The model applies a chemistry module named IFS(CB05) (Flemming et al., 2015) and an aerosol module as described by Morcrette et al. (2009). The model has 60 vertical levels between 0.1 hPa and 1000 hPa, and has a horizontal resolution of $0.75° \times 0.75°$ latitude $\times$ longitude. Output is provided every 3 h and includes a variety of meteorological parameters, concentrations of chemical tracers, and aerosol properties for

the time between 2003 and 2020. CAMS uses prescribed emissions from MACCity (MACC/CityZEN; Granier et al., 2011) for anthropogenic emissions, from MEGAN2.1 (Model of Emissions of Gases and Aerosols from Nature; Guenther et al., 2012) for biogenic emissions, and from GFAS v1.2 (Global Fire Assimilation System; Kaiser et al., 2012) for biomass burning emissions. GFAS assimilates information of fire radiative power and injection height from satellite observations. In a model

evaluation study, Wang et al. (2020) compared tropospheric trace gas profile measurements from aircraft campaigns to CAMS reanalysis data. They compared measurements taken over the Arctic, North America and Hawaii to model simulation results and show that simulated PAN is in agreement with observations, while CAMS $C_2H_6$ is generally underestimated. Further, Johansson et al. (2020) and Wetzel et al. (2021) compared GLORIA PAN, $C_2H_6$, and HCOOH to CAMS reanalysis data in the upper troposphere above the Asian Monsoon, and the North Atlantic, respectively. They found an overall underestimation

of those pollutants in the model, and for the Asian Monsoon it is suggested that some emission sources are missing in CAMS. Because these evaluations are limited to the northern hemisphere and do not capture large scale biomass burning events, our manuscript may contribute to estimates of the general performance of CAMS in reproducing pollution trace gas distributions in the upper troposphere.

### 2.3    HYSPLIT trajectory model

The HYbrid Single-Particle Lagrangian Integrated Trajectory model (HYSPLIT; Stein et al., 2015; Rolph et al., 2017) is a commonly used transport and dispersion model developed by the Air Resources Laboratory of the National Oceanic and Atmospheric Administration. In this work, we use HYSPLIT to calculate backward trajectories based on Global Forecast System (GFS) global model results with a horizontal resolution of $0.25° \times 0.25°$ latitude $\times$ longitude. Seven day backward trajectories are started at measurement geolocations with enhanced pollution trace gas signatures in the GLORIA measurements. Upward

transport of the trajectories is limited to the vertical motion prescribed by the meteorological fields. It is well known that these meteorological fields underestimate upward transport, in particular for convective events, which are usually not resolved or sufficiently parameterized by meteorological models (e.g., Hoyle et al., 2011). Advanced schemes for convection detection along backward trajectories (e.g., Wohltmann et al., 2019) are not applied for backward trajectories used by this study. For this reason, the vertical motion of the HYSPLIT trajectories is not discussed in detail here. Hence, we do not attempt to retrieve the

origin of the measured air masses, but rather the location, at which the air masses reached upper tropospheric altitudes.

### 3    Pollution trace gas measurements

### 3.1    Flight on 8 September 2019

The SouthTRAC research flight on 8 September 2019 was part of the transfer of the HALO aircraft from Oberpfaffenhofen, Germany to Rio Grande, Argentina and started in Sal, Cape Verde in southwest direction towards Buenos Aires, Argentina.

Figure 1a shows the flight track as well as PAN VMRs at the horizontal tangent point locations of the GLORIA field of view. To facilitate the interpretation of the trace gas cross sections in the following panels, measurement times are also indicated. Fire

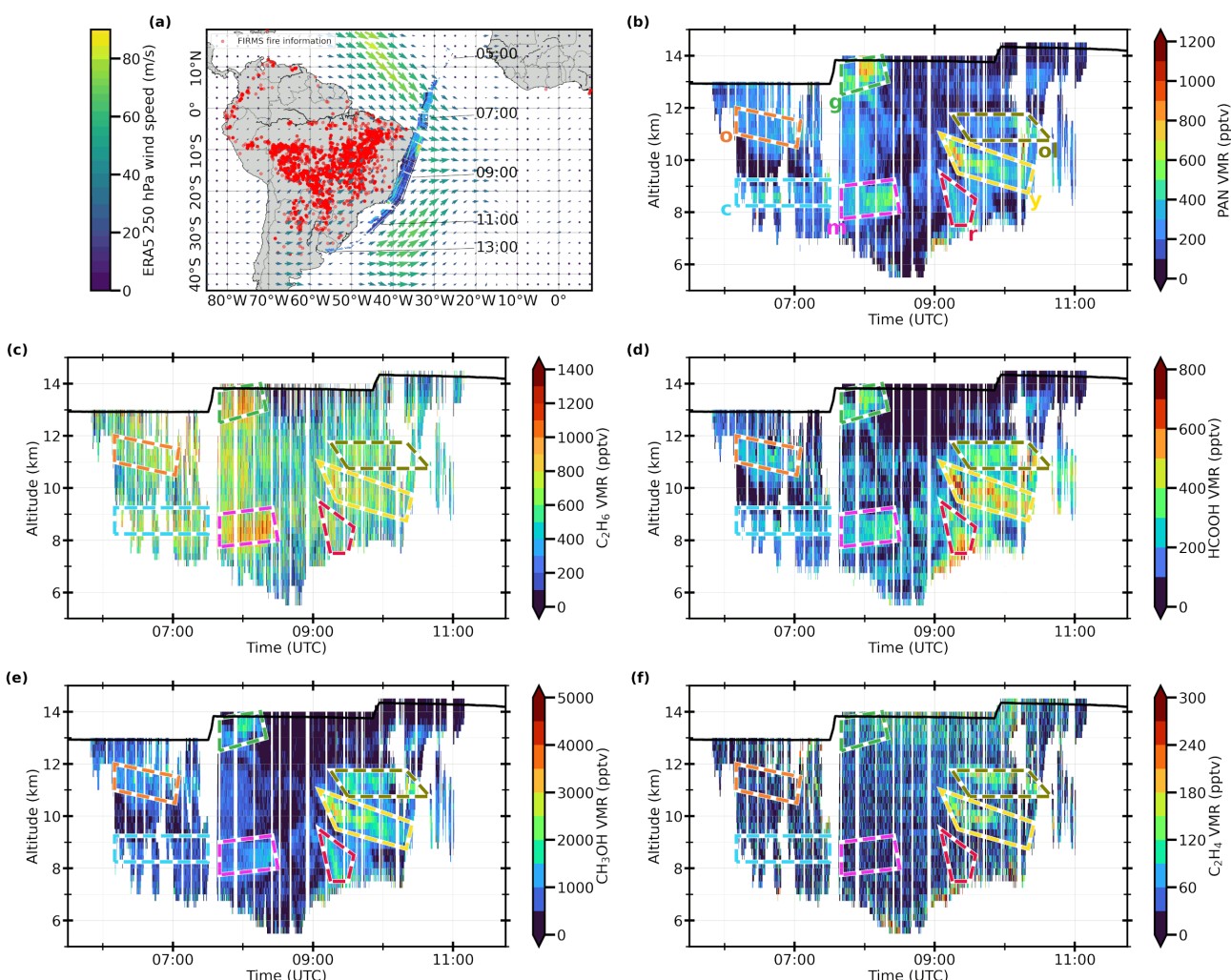

**Figure 1.** (a) Flight path for SouthTRAC research flight on 8 September 2019 from Sal, Cape Verde to Buenos Aires, Argentina. Along the flight track, concentrations of PAN are displayed at the horizontal distributions of the tangent points (colorbar in b). The temporal evolution of the flight is indicated by times in UTC, marked along the flight track. Fire events as noted by FIRMS during 3 days before the GLORIA measurements are indicated as red dots on the map. ERA5 horizontal winds at 8:00 UTC at a typical measurement altitude of 250 hPa (approximately 10 km) are shown as arrows with amplitudes given by the left colorbar. (b) Vertical distribution of GLORIA measurements of PAN for SouthTRAC flight on 8 September 2019. The black line marks the aircraft altitude. Colored boxes mark air masses of interest, which are discussed in more detail by a trajectory analysis in the following section (o: orange, c: cyan, g: green, m: magenta, ol: olive, y: yellow, r: red). (c) Same as (b), but for $C_2H_6$. (d) Same as (b), but for HCOOH. (e) Same as (b), but for $CH_3OH$. (f) Same as (b), but for $C_2H_4$.

events in South America and Africa during three days before the GLORIA measurements, obtained from the Fire Information for Resource Management System (FIRMS; see https://firms.modaps.eosdis.nasa.gov/, last access 17 January 2022) are shown

as red dots in the background of the map. These data are based on observations from the Moderate Resolution Imaging Spec-
troradiometer (MODIS; Giglio, 2000) onboard the NASA Terra and Aqua satellites and are shown here to provide an overview
of biomass burning events prior to the research flight. These fire data reveal a large density of fires in central South America.
Furthermore, horizontal winds at 250 hPa (approximately 10 km altitude) from the fifth generation of the ECMWF Reanalysis
(ERA5; Hersbach et al., 2018) are shown as colored arrows to provide meteorological context of the GLORIA measurements.
These horizontal winds indicate strong south-westerly currents reaching the flight path at the equator (approximately 7:00
UTC). Another region of strong winds is located above the South Atlantic with north-westerlies parallel to the flight path.
Above the South American continent, where many fire counts are displayed, weaker north-westerly winds are marked, which
may transport polluted air from the fires to the measurement locations. However, these winds, which are only shown at 250 hPa,
can only provide a simple overview of the situation, which will be more closely analysed by trajectory calculations below.

   The measured cross sections of the pollution trace gases reveal filamentary structures along the flight path. At the beginning
of the flight around 7:00 UTC, two layers of enhanced pollution trace gas concentrations are observed at altitudes of 8 km
and 11 km for all discussed trace gases, except $C_2H_4$ (marked by orange and cyan boxes). Compared to other enhancements
during this flight, these local maxima are relatively low (up to 500 pptv for PAN, 700 pptv for $C_2H_6$, 300 pptv for HCOOH,
and 1200 pptv for $CH_3OH$, Fig. 1b-e). During the later part of the flight, strong enhancements of PAN of up to 900 pptv are
measured at 8:00 UTC and 13 km altitude (green). These air masses also contain strongly enhanced $C_2H_6$ (up to 1000 pptv),
weaker enhanced HCOOH (up to 500 pptv) and $CH_3OH$ (up to 2000 pptv), but no signatures of enhanced $C_2H_4$. At the same
time, but at lower altitudes of around 8 km (magenta), a weaker local maximum is observed for PAN (up to 500 pptv), $C_2H_6$
(up to 1100 pptv), HCOOH (up to 400 pptv), and $CH_3OH$ (up to 1600 pptv), but again not for $C_2H_4$. Later during the flight,
at 9:15 UTC and 10 km altitude (yellow), a strong enhancement is measured for all five discussed trace gases (up to 1000 pptv
for PAN, 900 pptv for $C_2H_6$, 800 pptv for HCOOH, 4000 pptv for $CH_3OH$, and 200 pptv for $C_2H_4$). Above this horizontally
extending local maximum, a similar structure is observed at 11 km altitude (olive). For all measured trace gases but $C_2H_4$,
absolute VMRs are smaller compared to the enhancement at 10 km altitude. Again, at 9:15 UTC but at altitudes above cloud
tops (7 km; red), a local enhancement with smaller absolute VMRs is measured for all discussed trace gases, except $C_2H_4$.
Towards the end of the flight (after 10:15 UTC), measurements have been affected by aerosol or clouds, which resulted in
increased filtering of data and reduced retrieval quality (see Supplement).

   In horizontal-vertical space, the pollution gas distributions are strongly correlated, pointing to self-consistent measurements.
However, the highest VMRs for each trace gas are not always observed within the same air masses. For example, $C_2H_6$ shows
highest VMRs at the maximum at 8:00 UTC and 8 km altitude, but PAN, HCOOH, and $CH_3OH$ have comparably low absolute
VMRs for this local maximum. For $C_2H_4$, the species with the shortest atmospheric lifetime (see Tab. 1), only two maxima
are observed, in contrast to many distinct maxima in the other species. This indicates that these air masses have been recently
transported from a biomass burning source to the measurement location. A further investigation of the air mass history will be
presented in Sect. 4, based on the analysis of backward trajectories.

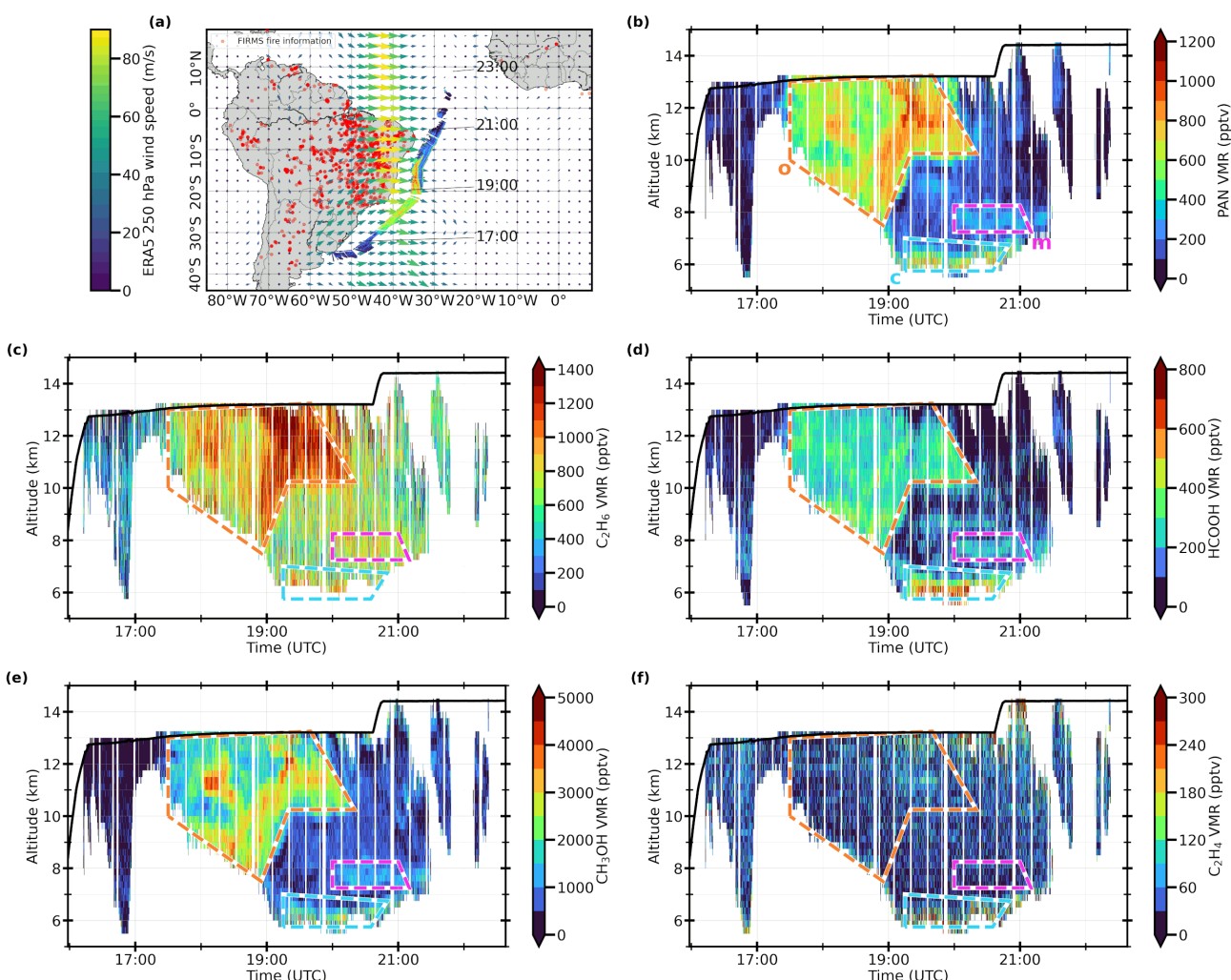

**Figure 2.** Same as Fig. 1, but for SouthTRAC flight on 7 October 2019 from Buenos Aires, Argentina to Sal, Cape Verde. In panel (a), ERA5 winds are displayed at 19:00 UTC. Note that colorbars may have changed compared to Fig. 1.

## 3.2 Flight on 7 October 2019

The flight on 7 October 2019 was similar to the flight on 8 September 2019, but in the opposite direction. The flight was routed from Buenos Aires to Sal, and this time, GLORIA tangent points were directed eastwards, away from the South American coast (see Fig. 2a). Again, a large number of fires was reported by FIRMS above central South America. The horizontal winds show a strong westerly component at approximately 40°W, which may have transported polluted air from the fires to the measurement locations.

Cross sections of the pollution trace gases for this flight are presented in Fig. 2b-f. Until 17:30 UTC, no considerable enhanced structure is observed for any discussed trace gas. Afterwards, a remarkable nose-shaped maximum is measured for

PAN, $C_2H_6$, HCOOH, and $CH_3OH$ between 17:30 UTC and 20:00 UTC, and 8 km to 13 km altitude (orange). Within this plume, maximum VMRs of PAN up to 1000 pptv, of $C_2H_6$ up to 1400 pptv, of HCOOH up to 500 pptv, and of $CH_3OH$ up to 4500 pptv are observed. For $C_2H_4$, only a small enhancement at 19:30 UTC and 11 km altitude with VMRs up to 120 pptv is measured. These are the same air masses, for which PAN and $CH_3OH$ have their highest VMRs within the larger plume. A smaller enhancement in all discussed trace gases is observed between 19:00 UTC and 21:00 UTC, above cloud top altitude, at 6 km (up to 800 pptv for PAN, 1100 pptv for $C_2H_6$, 800 pptv for HCOOH, 2500 pptv for $CH_3OH$, and 300 pptv for $C_2H_4$; cyan). However, these enhancements are affected by a considerably higher absolute total estimated error (see Supplement), and therefore these measurements are not as reliable as others. Further, a smaller maximum is measured between 20:00 UTC and 21:00 UTC at 7.5 km altitude (magenta) for PAN (300 pptv), $C_2H_6$ (900 pptv), HCOOH (300 pptv), and $CH_3OH$ (1500 pptv).

The ERA5 horizontal winds suggest that the largest plume observed during this flight was transported from fires in central South America. The relatively low measured VMRs for $C_2H_4$ indicate that the plume has been transported in the atmosphere for clearly more than a few days (the upper tropospheric lifetime of $C_2H_4$ is 1.2 days (see Tab. 1).

## 4   Trajectory analysis

For the geolocations (latitude, longitude, altitude, time) of the GLORIA measurements marked by colored boxes in the cross sections shown in Figs. 1-2 (colors are specified in Sect. 3), HYSPLIT seven day backward trajectories have been calculated and their horizontal evolution is presented in Fig. 3. For a comparison with potential emission sources, additionally surface PAN VMRs from the CAMS model are shown as seven day averages for the time before each flight. These HYSPLIT backward trajectories are not expected to reasonably reproduce vertical motion of the air masses and therefore no vertical information is presented in the following figures. For the interpretation of the trajectories, this means that the examined air mass may have reached UTLS altitudes at any point along the backward trajectory connected with this particular air mass.

Trajectories for the flight on 8 September 2019 are shown in Fig. 3a. Because of the many small scale maximum structures in the measurements, trajectories from many different colored regions have been calculated. It can be seen that cyan, magenta, and orange marked air masses come from central Africa, while green, red, yellow, and olive marked air masses come from South America. These different points at which the air masses reached UTLS altitudes are also in agreement with the measured distributions of $C_2H_4$, the pollution trace gas with the shortest atmospheric lifetime in this study. This gas is only enhanced within the yellow and olive marked air masses, which both are indicated to come from South America. However, the red and green marked air masses also come from South America (according to the HYSPLIT trajectories), but do not show enhanced VMRs of $C_2H_4$. In particular, trajectories for the red marked air masses look very similar to those marked yellow and olive. Possible reasons for these different $C_2H_4$ concentrations of air masses with similar history may be different emissions of $C_2H_4$, higher loss processes for the different air masses, or different timing of vertical transport, which is not analyzed here. In Fig. 3b, CAMS surface PAN VMRs, averaged one week before the flight, can be compared to the trajectory paths. CAMS surface PAN was shown to agree with measurements elsewhere (see Sect. 2.2 and  Wang et al., 2020). As expected from the FIRMS fire

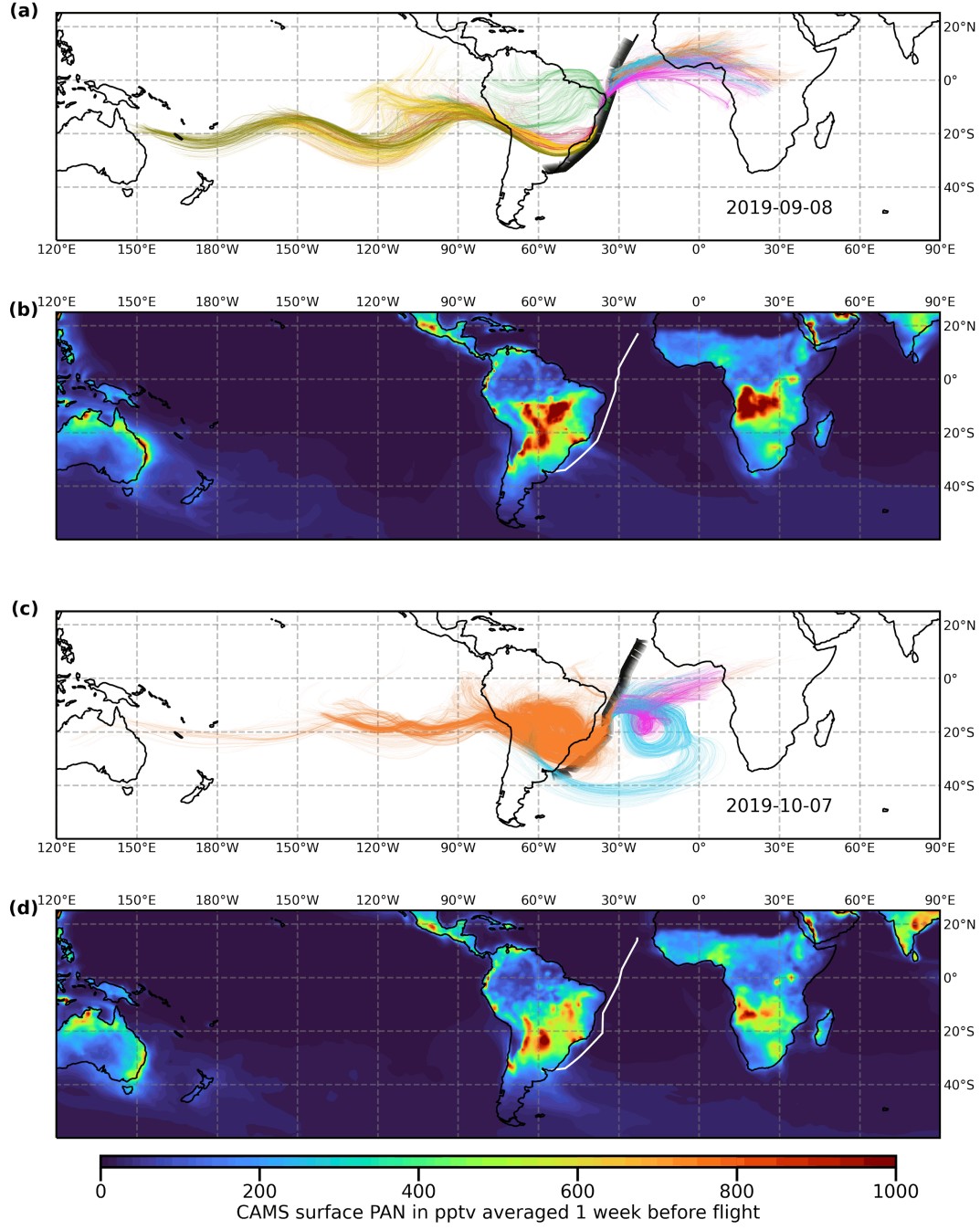

**Figure 3.** Horizontal evolution of HYSPLIT 7 days backward trajectories, starting at colored regions, as marked in Figs. 4-5 for flights on (a) 8 September 2019, and (c) 7 October 2019. The flight track and all tangent points are marked (see Figs. 4-5 for color bar) to indicate the beginning of the backward trajectories. Panels b and d show surface level volume mixing ratios of PAN, averaged over one week before the corresponding research flight, as reported by the CAMS model.

data shown in Fig. 1a, elevated PAN concentrations are present in central South America and central Africa. These are also the regions, where the HYSPLIT backward trajectories indicate the origin of the measured polluted air masses.

For the flight on 7 October 2019, three air masses of interest have been marked by colored boxes in Fig. 2. The paths of backward trajectories connected with these air masses are shown in Fig. 3c. As discussed in the previous section, pollution trace gas measurements of this flight are dominated by a major plume, marked with an orange box. HYSPLIT backward trajectories starting from this orange marked plume indicate an origin of these air masses from the central South American continent. As already estimated from the FIRMS fire data and ERA5 horizontal winds in Fig. 2a, it is likely that these polluted air masses come from biomass burning events in central South America. Surface concentrations of PAN (Fig. 3d), also illustrate enhancements in this region. Air masses from smaller maxima during this flight, marked cyan and magenta, instead have been first transported longer paths above the South Atlantic. The pollution trace gases of the cyan marked air masses have been emitted in South America, and then been transported above the South Atlantic for approximately one week, before they reached the measurement location. The backward trajectories associated with the magenta marked air masses also have been transported above the South Atlantic for one week before the measurement. Before this one week above the ocean, these air masses appear to come from central Africa, which is also a region of slightly enhanced PAN surface VMRs according to the CAMS model. These relatively lower enhancements of surface PAN are in agreement with the relatively lower enhancements of the GLORIA PAN VMRs.

## 5    Comparison to CAMS model simulations

In this section, GLORIA PAN, $C_2H_6$, HCOOH, $CH_3OH$, and $C_2H_4$ trace gas distributions will be used to evaluate CAMS model results. Figs. 4-5 present comparisons of these pollution trace gas measurements to CAMS simulation results, which have been temporally and spatially interpolated to the GLORIA tangent point locations and measurement times. In order to ensure a meaningful comparison, up to 27 GLORIA profiles are averaged horizontally to have a comparable horizontal resolution to the CAMS model ($\approx$80 km). In these figures, first direct comparisons of measured and simulated pollution trace gas cross sections are shown, followed by correlation plots. At the end of this section, implications for suggested model improvements based on these comparisons are discussed.

### 5.1    Flight on 8 September 2019

Figure 4 compares GLORIA measurements and CAMS simulation results for the research flight on 8 September 2019. For the direct comparison between distributions of PAN from GLORIA and CAMS (Fig. 4a-b), the model is able to reproduce all measured structures that are marked with colored boxes, except for the air mass marked in yellow. In addition, absolute maximum PAN VMRs are simulated too low, in particular for the green marked air masses. The correlation plots reflect this overall agreement between measurement and model, within the estimated errors of the GLORIA retrieval (see Tab. 2). In the point-wise correlation plot (Fig. 4c), it is illustrated that points that are far away from the dashed line are connected with yellow and green marked air masses, which have been identified as problematic in the cross section comparison before. With help of

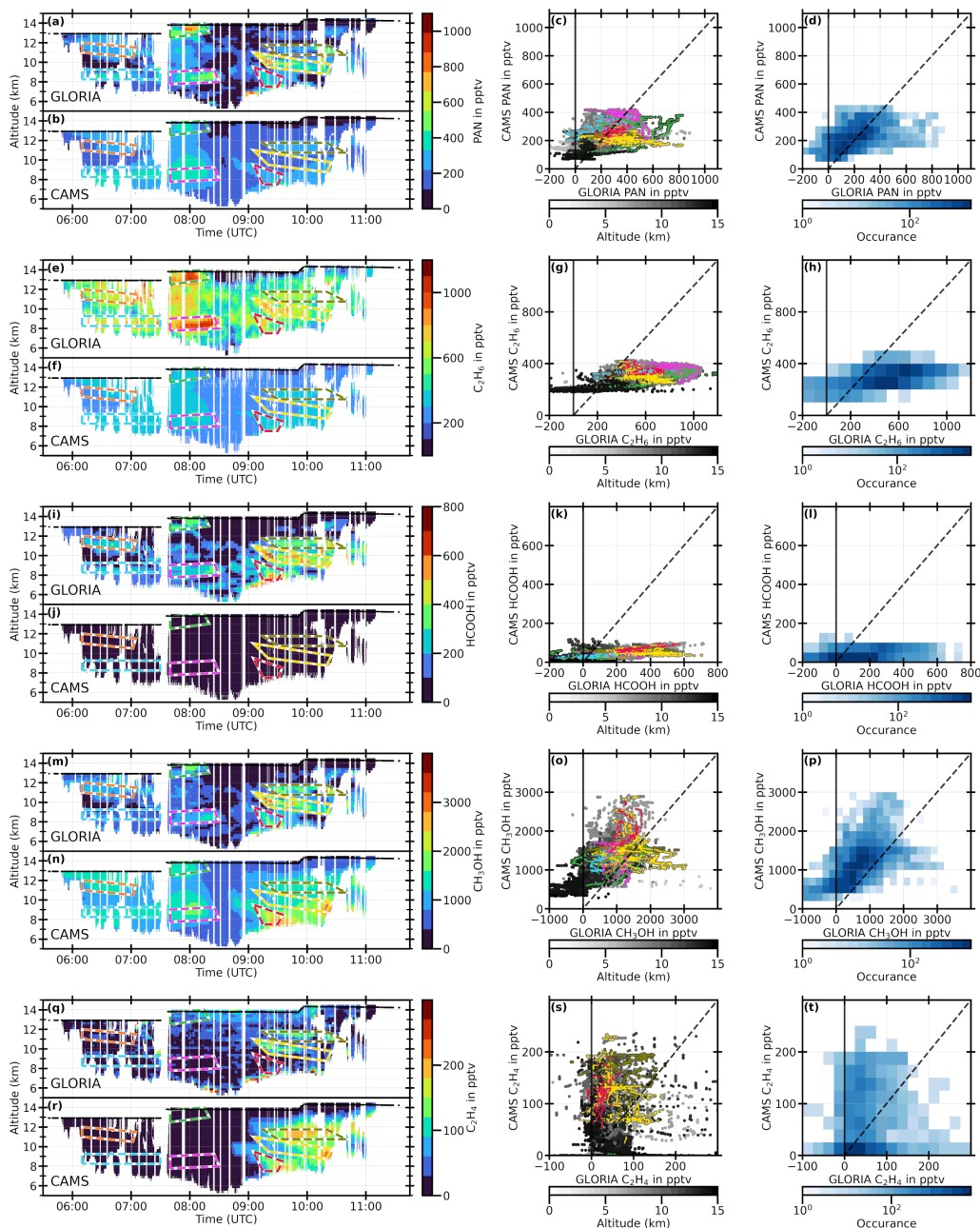

**Figure 4.** Comparison of GLORIA PAN (a-d), $C_2H_6$ (e-h), HCOOH (i-l), $C_2H_4$ (m-p), and $CH_3OH$ (q-t) measurements to interpolated CAMS simulation results during SouthTRAC flight on 8 September 2019. In the left column, measurement and interpolated model results are compared as cross sections. Colored boxes mark regions with measurements of enhanced pollution trace gases and are repeated from Fig. 1. Panels in the second column show single point correlations, color coded by altitude, while panels on the right display the occurrence of these correlation points in fixed bins.

the histogram plot (Fig. 4d) it is shown that the majority of correlation points is close to the dashed line (assuming a GLORIA total estimated error of up to 130 pptv).

In the direct comparison of $C_2H_6$ (Fig. 4e-f), some of the measured local maxima are reproduced by the model (e.g., red, magenta or green boxes), but overall $C_2H_6$ maxima and background values are underestimated by the CAMS model. This underestimation of $C_2H_6$ by the model is also illustrated by the correlation plots (Fig. 4g-h). The comparison of measured and simulated HCOOH is presented in Fig. 4i-l. While multiple enhancements up to 700 pptv are observed by GLORIA, CAMS simulates less than 100 pptv of HCOOH for the entire cross section.

In contrast to HCOOH, CAMS succeeds in reproducing the measured distribution of $CH_3OH$ qualitatively, as it does for PAN and $C_2H_6$. Relative maxima (except the one of the yellow marked air mass) are well represented by the model (Fig. 4m-n), and most of the absolute VMRs are in the same order of magnitude as the ones of the GLORIA measurements. However, background VMRs are typically simulated too high. The correlation plots (Fig. 4o-p) illustrate that the model overestimates $CH_3OH$ concentrations overall, in particular for the red marked air masses. For the missing simulated local maximum at the yellow marked air masses, CAMS underestimates measured VMRs.

For $C_2H_4$ (Fig. 4q-r), only distinct local maxima are observed by GLORIA during the second part of the flight, at the olive and yellow marked air masses. CAMS also only simulates enhancements of $C_2H_4$ during the second part of the flight, in accordance with the measurements. However, absolute $C_2H_4$ VMRs are simulated considerably higher than measured (typically 200 pptv compared to 150 pptv). In particular, the yellow marked air masses do not contain a simulated maximum, even though it was measured. In contrast, the red marked air masses have a simulated maximum, which was not measured. This overestimation is also shown in the correlation plots (Fig. 4s-t).

## 5.2 Flight on 7 October 2019

The flight on 7 October 2019 is characterized by a prominent nose-like plume. In Fig. 5a-b, the direct comparison of GLORIA and CAMS PAN is presented. The largest plume (marked with an orange box) is simulated by CAMS in quantitative agreement with the GLORIA measurements. A smaller measured local maximum structure above the cloud top (marked cyan), is also reproduced by CAMS, but with lower absolute VMRs (700 pptv measured compared to 300 pptv simulated). An even smaller measured local maximum (300 pptv) is marked by a magenta box. Although the absolute VMRs in this box are comparable in the simulation, this maximum is not distiguishable from the background VMRs, which are generally higher in the simulation than in the measurement. The overall agreement between measurement and simulation is confirmed by the correlation plots (Fig. 5c-d). As discussed, lower simulated VMRs for the cyan marked air masses stand out, together with overall slightly higher simulated PAN background VMRs.

For $C_2H_6$ (Fig. 5e-f), also the major plume (orange) is clearly visible in the GLORIA measurement. However, for the simulation, the plume's shape is not easy to recognize. A smaller enhancement is simulated within the orange marked air masses, and increasing up to 600 pptv (compared to more than 1400 pptv measured) until the middle of the flight. Secondary maxima that have been measured (magenta and cyan) are not reproduced by the model. This overall underestimation of CAMS $C_2H_6$ is also illustrated by the correlation plots (Fig. 5g-h).

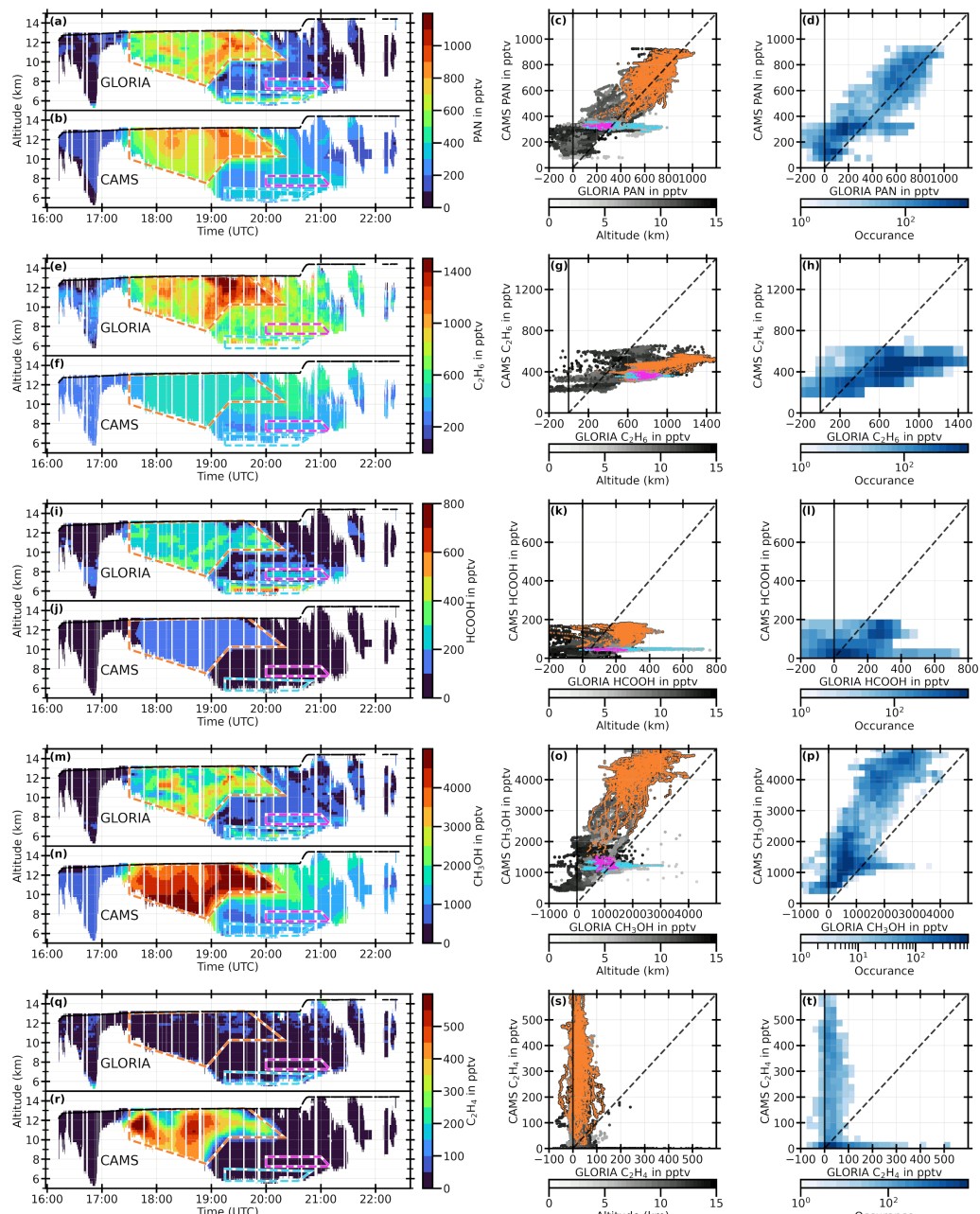

**Figure 5.** Same as Fig. 4, but for SouthTRAC flight on 7 October 2019. Note that colorbars may have changed compared to Fig. 4.

In Fig. 5i-j, comparisons of GLORIA and CAMS HCOOH are shown. The measurements again show the large plume (orange, up to 500 pptv) and a strong enhancement with a smaller extent above cloud top altitude (cyan, up to 800 pptv) and above (magenta, up to 300 pptv). However, CAMS is only able to reproduce the orange marked plume with considerably

smaller absolute VMRs, below 200 pptv, of HCOOH, while the cyan and magenta maxima are not reproduced at all. This underestimation in simulated HCOOH of the orange marked plume, and for the cyan marked air masses in particular is also visible in the correlation plots (Fig. 5k-l).

In contrast to $C_2H_6$ and HCOOH, $CH_3OH$ is overestimated by the CAMS model (Fig. 5m-n). The major plume (orange) is measured with up to 4000 pptv, while CAMS simulation results show more than 5000 pptv of $CH_3OH$. Secondary maxima (cyan and magenta) are measured for $CH_3OH$, but cannot be distinguished from the generally too high background of the simulation results. This is also reflected by the correlation plots (Fig. 5o-p), where the overall overestimation of $CH_3OH$ is illustrated, in particular for the orange marked plume. The cyan marked air masses are, however, slightly underestimated by CAMS.

For $C_2H_4$, only a tiny enhancement of less than 120 pptv is measured by GLORIA within the orange marked plume. In contrast, CAMS shows large enhancements of up to more than 600 pptv of $C_2H_4$ over the whole orange marked air masses (Fig. 5q-r). This overall overestimation is also reflected in the correlation plots (Fig. 5s-t).

## 5.3 Discussion

The comparisons between measured GLORIA pollution trace gas cross sections and CAMS simulation results show that CAMS is able to reproduce observed major plume structures. For PAN, absolute VMRs reveal a reasonable agreement, while $C_2H_6$ and HCOOH are underestimated, and $CH_3OH$ and $C_2H_4$ are overestimated by the model. The overall agreement for PAN in space and time even for most of the small scale structures measured during the flight on 8 September 2019 indicates the ability of the CAMS model to correctly transport the polluted air masses from reasonably assumed emission sources within the model. This strength of the CAMS model in reproducing structures of pollution trace gases correctly is also demonstrated in the comparison of the flight on 7 October 2019.

The underestimation of $C_2H_6$ and HCOOH, and the overestimation of $CH_3OH$ and $C_2H_4$ may be due to several reasons: The composition of polluted air masses is difficult to estimate a priori, leading to possibly erroneous trace gas emissions used by the model (Flemming et al., 2015). The GFAS emission inventory, which is used by CAMS for biomass burning emissions, assimilates fire radiative power from the MODIS satellite observations, and provides emissions for 40 gas phase and aerosol species according to emission factors from literature (Kaiser et al., 2012). This approach may result in a good representation of location and time of the emissions, but may also result in emission strengths that are better for some species than for others. Such emission factors could be improved by comparing them to other emission data sets, such as emissions, derived from satellite measurements (Stavrakou et al., 2012; Bauwens et al., 2016).

Further, atmospheric lifetimes may be influenced by uncertain or incomplete reaction properties of source and loss processes within the model. Missing or underestimated atmospheric loss processes may result in too large simulated VMRs, overestimated atmospheric loss processes may result in too low simulated VMRs. For HCOOH, it is known that a common model shortcut (transforming formaldehyde into HCOOH inside clouds without considering methanediol as intermediate) causes a too much washout of HCOOH and too low HCOOH concentrations overall (Franco et al., 2021).

## 6 Conclusions

This study discusses simultaneous airborne measurements of PAN, $C_2H_6$, HCOOH, $CH_3OH$, and $C_2H_4$ measured by GLORIA during the SouthTRAC campaign in September and October 2019. Both research flights discussed have been performed above the South Atlantic, and revealed different two dimensional vertically resolved distributions of the multiple pollution trace gases. While the flight on 8 September 2019 revealed a filamentary structure of pollution trace gas enhancements, the flight on 7 October 2019 was characterized by a large plume with high absolute VMRs of all discussed trace gases but $C_2H_4$. For each flight, two dimensional distributions of the discussed trace gases are coherent, considering the different atmospheric lifetimes, showing the good quality of the GLORIA measurements.

The analysis of the horizontal components of HYSPLIT backward trajectories starting at GLORIA measurement geolocations with enhanced pollution trace gas VMRs illustrate that these upper tropospheric air masses measured above the South Atlantic reached UTLS altitudes above South America and Africa. Although on the flight on 8 September 2019 polluted air masses showed a filamentary structure and came from both, South America and central Africa, the second flight discussed was characterized by a major plume, coming from central South America. Smaller polluted air masses measured during that flight traveled above the South Atlantic for approximately one week after they were released either from South America or Africa. This trajectory analysis also helps to explain the enhancements of $C_2H_4$, which only were observed in some of the places where peaks in other measured pollution trace gases were found. From the trajectory analysis, it is confirmed that these air masses with the shorter lived substance $C_2H_4$ enhanced, have been recently released in South America and have not traveled for long distances.

The comparison of the GLORIA cross sections with interpolated CAMS reanalysis data illustrates the strength of the CAMS model to reproduce measured PAN VMR distributions. For PAN, structures and absolute VMRs are repeated by the model as expected for the given model resolution. This is in line with results from Wang et al. (2020) for the northern hemisphere. Structures of CAMS $C_2H_6$ are overall in agreement with GLORIA for both discussed research flights, but absolute VMRs are underestimated. This underestimation of $C_2H_6$ has also been observed in the northern hemisphere by Wang et al. (2020). HCOOH is also largely underestimated by CAMS, which has been observed by Wetzel et al. (2021) in the northern hemisphere, too. During the flight on 8 September 2019, enhancements of the model HCOOH data are so low that it is not possible to distinguish those from typical background VMRs. Simulations for the flight on 7 October 2019 are able to reproduce the large plume of HCOOH qualitatively, but with considerably too low absolute VMRs. As discussed, this underestimation of HCOOH by the model is possibly caused by a shortcut of HCOOH chemistry in clouds (see Franco et al., 2021). $CH_3OH$ instead is overestimated by the model, for both, peak and background VMRs. Structures measured by GLORIA are, however, reproduced by CAMS. This indicates that surface emission locations are simulated correctly, but emission strength might be overestimated, or a missing or underestimated atmospheric sink may cause this simulated overestimation in $CH_3OH$. Such a missing atmospheric sink would also influence its estimated atmospheric lifetime. For $C_2H_4$, CAMS is able to simulate the measured filaments of the flight on 8 September 2019, but overestimates the absolute VMR and shows an enhancement at air masses that do not have observed elevated VRMs. CAMS correctly shows no enhancement of $C_2H_4$ for parts of the flight,

where GLORIA measurements do not indicate enhancements. This is in contrast to the flight on 7 October 2019, where CAMS showed strong enhancements within the large plume, that has been measured for all trace gases but $C_2H_4$. Overall, CAMS also overestimates absolute VMRs of $C_2H_4$.

In summary, we find that PAN is simulated remarkably well by the model, and that for the other discussed trace gases measured structures are reproduced well overall. This indicates that the location of the emission sources and the atmospheric dynamics of the CAMS model perform well. For PAN, the emission strengths of precursor gases, and atmospheric gain and loss processes of the model enable CAMS to reproduce observed distributions that quantitatively compare with the GLORIA measurements. However, for the other discussed pollution trace gases, absolute VMRs are not in agreement with the GLORIA observations. It is suggested that emissions strengths and atmospheric source and loss processes for these species could be improved in CAMS. In particular, for HCOOH, known problems should be avoided by not using the common model shortcut. Further, emission fluxes from the GFAS emission inventory, which is used by CAMS, could be compared to other emission data sets to improve emission factors within GFAS for $C_2H_6$, HCOOH, $C_2H_4$, and $CH_3OH$. Our comparison with CAMS shows the strength of high resolution altitude resolved measurements of multiple species in order to find strengths and deficiencies of atmospheric models.

*Data availability.* GLORIA measurements are available in the database HALO-DB (https://halo-db.pa.op.dlr.de/mission/116, last access 20 January 2022) and on the KITopen repository (Johansson et al., 2022). The CAMS model data is available from the Copernicus Atmosphere Data Store (https://ads.atmosphere.copernicus.eu, last access 20 January 2022).

*Author contributions.* SJ initiated the study, performed the analyses, and wrote the manuscript. GW, MH, JU, AK, NG, and SJ performed the GLORIA data processing. FFV, TN and coworkers operated GLORIA during the SouthTRAC campaign. BMS and coworkers coordinated the SouthTRAC campaign. All authors commented on and improved the manuscript.

*Competing interests.* The authors declare that they have no conflict of interest.

*Acknowledgements.* We gratefully thank the SouthTRAC coordination team, in particular DLR-FX for successfully conducting the field campaign. The results are based on the efforts of all members of the GLORIA team, including the technology institutes ZEA-1 and ZEA-2 at Forschungszentrum Jülich and the Institute for Data Processing and Electronics at the Karlsruhe Institute of Technology. We thank ECMWF for providing CAMS data. The authors acknowledge the NOAA Air Resources Laboratory (ARL) for the provision of the HYSPLIT transport and dispersion model used in this publication. We acknowledge support by the German Research Foundation (Deutsche Forschungsgemeinschaft, DFG Priority Program SPP 1294). We further acknowledge support by the KIT-Publication Fund of the Karlsruhe Institute of Technology.

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
