# Peer review of "Biomass burning pollution in the South Atlantic upper troposphere: GLORIA trace gas observations and evaluation of the CAMS model"

_Atmospheric Chemistry and Physics, 2021_

## Author Comment (AC1)

**Answer to Referee report #1**

We thank the referee for valuable comments and suggestions. In particular, we appreciate the efforts of the referee to polish the language of our manuscript. Our answers are given below. The original referee comment is repeated in **bold**, changes in the manuscript text are printed in *italics*.

**Overall comments**
* * *
**This is a very nice paper describing a well-executed solid piece of research that exploits a very new highly innovative set of observations, compares them to a state-of-the art atmospheric chemistry modeling system, and draws useful conclusions that have the potential to improve the accuracy of such models in future. I believe this paper is in very good shape and happily recommend it for publication in ACP once a few minor comments of mine (mostly related to wording) are considered (along with those of other reviewers, naturally). The standard of the presentation is excellent and the description of the work is laudably clear and well constructed.**

We thank the referee for this encouraging judgement.

**Minor comments**
* * *
**Line 10: Suggest: "...pollutants likely originate from..."**
Done.

**Line 12: Suggest: "...In comparisons to results of the CAMS..." (think "simulations" disrupts the flow and is not needed).**
Agreed.

**Line 14: Suggest you delete "too".**
Agreed.

**Line 16: Move "discussed" to after "gases"**
Done.

**Line 23: "their" is a vague antecedent. Does it refer to Africa, South America, etc., or to the increase in biomass burning, or to the UTLS. Also the "too" (end of the line) is out of place as the preceding sentence does not talk about any "increase" just the ongoing importance of the UTLS. I suggest you re think these two sentences a bit.**
We rephrased the second part of this sentence: *Due to increasing biomass burning activities in Africa, South America and Australia (Torres et al., 2010, Abram et al., 2021), the potential influence of biomass burning trace gases on climate may increase over time.*

**Line 29: Suggest: "Typical biomass burning trace gases have different atmospheric lifetimes and atmospheric sinks. Further, they may have additional non-pyrogenic**

**sources."**
Thank you for this excellent suggestion.

**Line 36: Suggest: "Atmospheric model simulation of such pollution trace gases is challenging:"**
Agreed.

**Line 42: Suggest "to simulate" -> "the accurate simulation of", then "due to" -> "during".**
Done.

**Line 43: Suggest: "processes, which" -> "processes that".**
Done.

**Line 44: Delete comma after gases, change "like" to "such as". Change "For example" to "As an example" (the latter somehow feels better as the example refers to more than just the preceding sentence).**
Thanks.

**Line 38: Delete "Then" (and capitalize "The" as a result, naturally).**
We think this was related to line 58. We changed it as suggested on this line.

**Table 2: Would be good to specify whether the estimated error is random/precision-like or systematic/accuracy-like, or some combination of the two.**
We changed the table caption to: *[…] estimated errors (combination of random and systematic errors).*

**Line 105: "this" -> "that"**
Done.

**Line 106: "estimate the" -> "estimates of the"**
Done.

**Figure 1 (and 2): I find the colored background (land/sea) in panel a distracting. I'm usually in favor of such things, but I think in this case there is simply too much else going on. I suggest you revert to a simple outlines as in Figure 3, or perhaps just shade the continents in pale grey.**
We agree with the referee and changed the map background for both figures.

**Figure 1, also: I find the green contour particularly hard to spot (compared to the others). Perhaps, for this one, you could put a thicker white contour underneath it to make it stand out (or perhaps used a dashed green/white line for the counter, but that might make it stand out too much).**
We increased all white counter lines in width and changed the line style to dashed lines for all colored boxes. We applied those changes to Figs. 4-5, too.

**Figure 1/2: I imagine you considered this, but it might be preferable to use the same color bar ranges between the two plots. They're not that different (in some cases they're already identical), and it might make for easier comparisons. On the other hand, if your feeling is that it mutes the enhancements in Figure 1 too much, I'm fine if you opt to keep**

**things as they are.**
We agree that the differences in the color bars have been very small and, in fact, the enhancements in Fig.1 are still well visible with the color bar ranges of Fig. 2 applied. **Line 153: Suggest you add a comma between "correlated" and "pointing"**
Done.

**Line 183: Suggest: "and their horizontal evolution is presented in Fig. 3"**
Agreed.

**Line 183/184: Suggest you swap "PAN" and "surface"**
Done.

**Line 184: Suggest "of" -> "from"**
Done.

**Line 186: My only semi-substantive point here. I'd avoid "origin". To me the "origin" is the place where the air masses left the surface, which is not constrained to lie along the trajectories. Rather I'd say "the point at which the air masses were injected into the UTLS", "or reached UTLS altitudes" or something like that.**
We agree with the referee and we now avoid the word "origin" throughout the manuscript, unless "origin" is used in a rather vague way (e.g.: "indicate the origin").

**Line 206/207: Suggest: "...also illustrate enhancements in this region."**
Agreed.

**Line 230: Delete "correlation", change "is" to "are"**
"Correlation" deleted as suggested. In our opinion, "is" refers to "the majority" and should be singular (and not plural "are" as suggested). But we are no native speakers – so we may be wrong, and in case of publication in ACP, copy editing will fix it.

**Line 233: Suggest "simulated too low" be changed to "underestimated" (even though you have "underestimation" in the next line, I think this is still preferable.**
Agreed.

**Line 237: Suggest: "to reproduce" -> "in reproducing"**
Done.

**Line 238: Suggest you insert "well" before "represented" (feel some word if needed, I'm fine with "fairly well" if you prefer)**
Done.

**Line 240: "In the" -> "The", "it is illustrated" -> "illustrate". Move "overall" to after "concentrations" on the next line.**
Done.

**Line 245: "accordingly" -> "in accordance"**
Done.

**Line 265: "large" -> "strong", "of" -> "with a"**
Done.

**Line 267: "too low" -> "smaller".  Add commas before and after "below 200 pptv"**
Done.

**Line 268: "of" -> "in"**
Done.

**Line 285: "to reproduce" -> "in reproducing"**
Agreed.

**Line 287: "have" -> "be due to"**
Done.

**Line 288: Perhaps add "a priori" or something after "estimate", to indicate that you're not talking about the composition measured by GLORIA, but rather in the immediate vicinity of the fires.**
Thank you for this suggestion.

**Line 291: Suggest "high" -> "much"**
Done.

**Line 291/292:  Move "overall" to after "concentrations"**
Done.

**Line 295: Move "discussed" to after "flights".  Change "above" to "over"**
Done.

**Line 296: Change "discussed to "multiple".**
Done.

**Line 300: suggest "retrieval" -> "measurements" (it's not just the retrieval that's doing well)**
Agreed.

**Line 302: "illustrated" -> "illustrate"**
Done.

**Line 303: "While during" -> "Although on",**
Done.

**Line 305: Delete "have"**
Done.

**Line 306: Delete the comma after "week" and change "have been" to "were"**
Done.

**Line 307-308: Suggest: "... which only were observed in some of the places where peaks in other measured pollution trace gases were found."**
Agreed.

**Line 327: Suggest "... are reproduced well overall..."**
Done.

**Line 328: "also" -> "the"**
Done.

**Line 329: "are able" -> "enable CAMS to"**
Done.

**Line 321: Change "should" to "could" and move "for these species" to just before "could"**
Done.

**References:  Do you really need both the doi and Copernicus URLs? (More of a question for the copy-editor).**
For previous publications in ACP, this was requested during the typesetting process. We suggest to leave the references as they are, if not requested otherwise by the copy-editor.

---

## Author Comment (AC2)

**Answer to Referee report #2**

We thank the referee for valuable comments and suggestions. Our answers are given below. The original referee comment is repeated in **bold**, changes in the manuscript text are printed in *italics*.

**In this study, the authors presented detailed analyses of five chemical species (PAN, $C_2H_6$, HCOOH, $CH_3OH$ and $C_2H_4$) measured by the Gimballed Limb Observer for Radiance Imaging of the Atmosphere (GLORIA) instrument during the Transport and Composition in the Southern Hemisphere Upper Troposphere/Lower Stratosphere campaign (SouthTRAC) conducted in over the South Atlantic in September-October 2019. In addition to the in-situ measurements, a back trajectory model (HYSPLIT) is used to examine the origins of the pollutants. The Copernicus Atmosphere Monitoring Service (CAMS) model simulations are also used to examine the transport pathways. The enhancements in those five chemical species, which were captured during each flight were found to have varying degree of agreement with the CAMS model results. This study presents a compelling result by utilizing a valuable set of data and the global and trajectory models. I would like to suggest a few minor changes which might add richness to this work.**

We thank the referee for the encouraging statement.

**General Comments:**

- **I would like to suggest adding a little more background on the five chemical species chosen in this work. What do they have in common? Why were those selected? How much understanding do the community has in terms of their sources, sinks and their chemical lifetime?**
  We thank the referee for this suggestion. In the introduction, we added a statement about the selection and the commonalities of these specific species (see answer to the specific point below). Further, we now discuss sources, sinks and lifetimes (which are all presented in Tab. 1) in the introduction, too.

- **Adding some information about the measurements of those species by satellites would be helpful, if possible. Are there any references comparing the satellite measurements and the model simulations? Do other models have difficulty simulating those species accurately? Adding a few relevant references would help understanding the general aspect of those species.**
  In the introduction, we mention satellite measurements in nadir and limb geometry for the discussed species. We now extended this part of the introduction with examples for usage of satellite data in modelling studies.

- **Does the CAMS model perform well in general? I would like to see a statement about why the CAMS model is used here. Is the goal to evaluate the model or to improve the model? If the improvement is the goal, a more specific direction would be needed possibly in conclusion.**
  In the discussion of Sec. 5, and in the conclusions, we write that CAMS performs well for the species PAN, and we address issues with the other trace gases, together with educated guesses why these other trace gases perform not as good as PAN.

In the introduction, we added a statement why the CAMS model is chosen for this study, and the goal of this study: *The CAMS reanalysis uses a state-of-the-art atmospheric chemistry model for data assimilation, which is publicly available and widely used for air quality and pollution related studies (e.g., studies citing Inness et al., 2019). In this work, we aim to evaluate the CAMS reanalysis in the remote upper troposphere above the South Atlantic, a sparsely measured region. With our comparisons we further aim to give recommendations for improving the CAMS model with respect to emissions and atmospheric lifetimes for the studied species.*

- **It is stated throughout the study that the degree of agreement between the measurements and the model varies depending on the species. I would like to suggest adding more thoughts or references to make the findings valuable. If the agreement is not good, how can we improve it in the future?**
  We extended the relevant parts of the manuscript according to the specific points raised by the referee below. For the improvement of CAMS, we make suggestions (variation of emission strengths in the emission inventory, review of atmospheric gain or loss processes).

**Specific Comments:**

**P1, L14: Are PAN, $C_2H_6$ and HCOOH longer-lived than $CH_3OH$ and $C_2H_4$? I am curious why the agreement between the measurements and the model is better for PAN only.**
As shown in Tab. 1 later in the manuscript, PAN, $C_2H_6$ and HCOOH have upper tropospheric lifetimes longer than weeks, while $CH_3OH$ and $C_2H_4$ rather have lifetimes of few days. As explanation why the agreement between the measurements and the model is better for PAN only, we suggest later in the abstract *model deficiencies in the representation of loss processes and emission strength.*

**P2, L23 & 24: I recommend listing examples of 'some of these traces gases' and 'some pollution trace gases' here.**
We added carbon monoxide and nitrogen dioxide as examples for ozone precursor biomass burning gases, together with additional references (Bozem et al., 2017; Bourgeois et al., 2021). Further, we name now VOCs for aerosol formation, because the contribution of single trace gases to secondary aerosol formation are still subject to current research. Again, we give additional references (Hobbs et al., 2003; Akherati et al. ,2020).

**P2, L28: It would be helpful to add a reference at the end of this sentence or rephrase this as 'their potential influence on climate may increase over time'.**
We adapted your suggested change together with suggestions from the other referee.

**P2, L29: I recommend making changes to this sentence. For instance, 'and may have other sources in addition to pyrogenic emissions.**
We changed this sentence according to the other referee's suggestion.

**P2, L30: Why those five gases were chosen and what do they have in common?**
We added to the manuscript: *These trace gases have been selected for this study, because they all are potentially emitted from biomass burning events, because they have a large range of upper tropospheric lifetimes (from a few days to several months), and*

*because they are part of the GLORIA (Gimballed Limb Observer for Radiance Imaging of the Atmosphere) measurements and of the CAMS (Copernicus Atmosphere Monitoring Service) model output. In addition, these trace gases are measured by various infra red satellite sounders (see below) but in coarser spatial resolution than the GLORIA measurements.*

**P2, L36: 'Filamentary structures' have been mentioned throughout the manuscript. It would be helpful to have a definition or description of it.**
We clarified that we mean mesoscale structures with horizontal extension of up to hundreds of kilometers by the term 'filamentary structures'.

**P2, L37: I recommend modifying the sentence 'Biomass burning events are typically represented by emission data sets in atmospheric models'. I think emission inventories are one of the factors determining how the model represents the biomass burning events. In fact, emissions, chemistry, and transport all make contributions to the model performance.**
We clarified the beginning of this paragraph: *Atmospheric model simulation of such pollution trace gases is challenging: For good model performance, knowledge about pollutant emissions, chemistry and transport are necessary. Location, time and emitted species of biomass burning events are typically represented by emission data sets in atmospheric models.*

**P2, L43: Adding more explanation about 'atmospheric processes' would be useful here. Does this refer to a chemical reaction or a physical process?**
We changed 'atmospheric processes' to 'chemical reactions and physical processes' and give now an additional example.

**P2, L48: Is there a website or a reference for the SouthTRAC campaign?**
We added a link to the SouthTRAC website. An overview paper is only available for the gravity wave part of the campaign (which is not the focus of this paper). This reference (Rapp et al., 2021) is cited in Section 2.1, where the flight campaign is introduced in more detail.

**P6, L118: Does this mean that only the horizontal motions will be analyzed here? Can we still trust the horizontal motions from the trajectories when the vertical motion is not accurate?**
Now, we clarify: *For this reason, the vertical motion of the HYSPLIT trajectories is not discussed in detail here , and it is not tried to retrieve the origin of the measured air masses, but rather the location, at which the air masses reached upper tropospheric altitudes.*
The original formulation that the horizontal motions are not analyzed may have been misleading. However, the vertical motion is limited by the meteorological fields, as mentioned earlier in this section. In particular, fast upward transport may be not represented in the meteorological fields. This means, that the air masses of the trajectories may have entered upper tropospheric altitudes at any point along the trajectory. In section 4, we discuss the trajectories very carefully due to this limitation in the vertical transport. According to the comment of referee 1, we further now avoid the formulation "origin of air masses" and rather speak of the "location at which the air mass reached the upper troposphere".

**P6, Section 3.1: It would be necessary to include references for FIRMS, MODIS and ERA5 in this section.**

We added the FIRMS website, and these references for MODIS and ERA5:
* Giglio, L.: MODIS/Aqua+Terra Thermal Anomalies/Fire locations 1km FIRMS V006 and V0061 (Vector data), 10.5067/FIRMS/MODIS/MCD14ML, NASA EarthData, 2000.
* Hersbach, H., Bell, B., Berrisford, P., Biavati, G., Horányi, A., Muñoz Sabater, J., Nicolas, J., Peubey, C., Radu, R., Rozum, I., Schepers, D., Simmons, A., Soci, C., Dee, D., and Thépaut, J.-N.: ERA5 hourly data on pressure levels from 1979 to present., 10.24381/cds.bd0915c6, Copernicus Climate Change Service (C3S) Climate Data Store (CDS), 2018.

**P7, Figure 1: This is a very nice set of figures. However, the boxes with various colors make the figure a bit complicated. It would be helpful to add the names of the gases where the maximum exists. For instance, add 'C2H6' in the pink box in Fig. 1c. This can also be considered for Fig. 2.**
We thank the referee for this suggestion, but we are not sure how to adapt this suggestion into the figures. For the magenta box, it would be easy to call it the "$C_2H_6$ box", but for the yellow box, it could be called "PAN box", "HCOOH box" or "$CH_3OH$ box" as all of those gases have strong maxima within this yellow box. In addition, there are more boxes than presented trace gases, so it is not possible to find a unique name for each box, if you are limited to the names of the trace gases.

**P10, Figs. 3a & 3c: It would be useful to mark the initialization locations in these plots. For instance, add larger dots on the location with the same color with the trajectories.**
We agree with the referee that it is not easy to see where the trajectories start. Unfortunately, marking the starting points with larger dots in the same color does not help to make the already busy plot easier to read. Instead we added all tangent points along the flight track, color coded with the grey scale color bar from Figs. 4-5 and changed the figure caption accordingly.

**P11, L197: Have there been any studies showing the CAMS performance on simulating PAN?**
As noted in Sec. 2.2, Wang et al. (2020) compared CAMS reanalysis PAN to aircraft measurements over the Arctic, North America and Hawaii and found an agreement between model and measurements. For higher altitudes, Wetzel et al. (2021) and Johansson et al. (2020) indicate an underestimation of PAN by the model above the North Atlantic, and within the Asian Monsoon, respectively. Further, within the Asian Monsoon, it is suggested that emission sources are missing. However, for CAMS surface PAN, the Wang et al. (2020) reference indicates a good performance.
We add those two references (Wetzel et al. (2021) and Johansson et al. (2020)) to Sec. 2.2, and add to Sect. 4:
*CAMS surface PAN was shown to agree with measurements elsewhere (see Sect. 2.2 and Wang et al., 2020).*

**P13, Section 5.1: It would be helpful to add some insights on the different degree of agreement between the measurements and the model depending on each species. Is it related to lifetime of the species? Or surface emissions? Why does the model overestimate CH₃OH?**
We now refer in the beginning of section 5 to the end of the section, where the different degree of agreement between measurements and model depending on the species are

discussed. Further, we extended this discussion subsection, according to the comment below.

**P15, L287: This is one of the most important findings in this work. I would recommend spending more time on the discussion. Are the sources of C$_2$H$_6$ and HCOOH underestimated in the models and well known? If CH$_3$OH and C$_2$H$_4$ are overestimated in the model, could that be related to the surface emissions only? A few references on this subject might be useful to include here.**
We extended the discussion as recommended by the referee. Now, we explain in more detail the influence of emission strength from the GFAS emission inventory, and we discuss the influence of too weak or too strong atmospheric loss processes to simulated VMRs.

**P16, L314: Does 'which has been also observed' refer to the underestimation of C$_2$H$_6$ in the Northern Hemisphere as well?**
We split this sentence into two, in order to make it clearer: *Structures of CAMS C$_2$H$_6$ are overall in agreement with GLORIA for both discussed research flights, but absolute VMRs are underestimated. This underestimation of C$_2$H$_6$ has also been observed in the northern hemisphere by Wang et al. (2020).*

**P16, L319: It would be helpful to add a sentence after this. Could this overestimation be related to overestimation of surface emissions or missing sink reactions? Or could this mean that the lifetime estimation is inaccurate?**
We extended this part according to the referee's suggestion: *CH$_3$OH instead is overestimated by the model, for both, peak and background VMRs. Structures measured by GLORIA are, however, reproduced by CAMS. This indicates that surface emission locations are simulated correctly, but emission strength might be overestimated, or a missing or underestimated atmospheric sink may cause this simulated overestimation in CH$_3$OH. Such missing atmospheric sink would also influence the estimated atmospheric lifetime.*

**P16, L326: This paragraph discusses a very important point. I would recommend adding a bit more specific information about the emission inventories. For instance, adding a few different emission inventories and discuss how they underestimate or overestimate specific species might give clearer idea about the future improvements. The current paragraph discusses this issue as a general issue but not specific to this study.**
We have extended the discussion part of Sect. 5, according to a previous comment, to discuss the influence of the emission inventory in more detail. Further, we extended the conclusions, according to the referee comment.